# Nursing students and nurses' knowledge and attitudes regarding children's pain: A comparative cross-sectional study

**Abigail Kusi Amponsah**[1,2☯]*, **Evans Oduro**[2☯], **Victoria Bam**[2‡], **Joana Kyei-Dompim**[2☯], **Collins Kwadwo Ahoto**[2☯], **Anna Axelin**[1‡]

**1** Department of Nursing Sciences, Faculty of Medicine, University of Turku, Turku, Finland, **2** Department of Nursing, Faculty of Allied Health Sciences, Kwame Nkrumah University of Science and Technology, Kumasi, Ghana

☯ These authors contributed equally to this work.
‡ These authors also contributed equally to this work.
* abkuam@utu.fi, akamponsah.fahs@knust.edu.gh, abacious@live.co.uk

**Data Availability Statement:** All relevant data are within the manuscript and its Supporting Information files.

## Abstract

### Introduction

Nurses encounter children who report of pain of diverse and unknown causes in their professional work. The current study therefore assessed and compared nursing students and nurses' knowledge and attitudes pertaining to children's pain in the Ghanaian context. The goal of this was to have a baseline information to guide the development and implementation of the content for a sustainable educational programme (short-course) for nursing students and nurses in Ghana.

### Methods

Between October and December 2018, a cross-sectional study was carried out among 554 final year nursing students and 65 nurses in Ghana. The Pediatric Nurses Knowledge and Attitudes Survey Regarding Pain (PNKAS) was used to collect data from participants who were affiliated to four educational institutions and eight hospitals. Data were descriptively and inferentially analyzed using chi-square test of independence, independent samples t-test and one-way analysis of variance (ANOVA).

### Results

Our findings revealed that nursing students and nurses generally had unsatisfactory knowledge and attitudes towards pain management in children. Nursing students however, had significantly higher scores than nurses in the total PNKAS score and in 10 out of the 13 identified item-areas. Greater scores were obtained by nursing students in areas which were related to pain physiology, pharmacokinetics, pharmacology of analgesics and pain perceptions (p < .05). All the participating nurses could not accurately determine: the onset of action of orally administered analgesics, equianalgesia of orally administered morphine, and

**Funding:** The research reported in this publication was partly funded by the researchers themselves and the Faculty of Allied Health Sciences, Kwame Nkrumah University of Science and Technology, Ghana to AKA. The content was produced by the researchers and does not in any way represent the official view of the funder. The funder had no role in any of the research processes from the conceptualization stage to the manuscript preparation phase.

**Competing interests:** The authors have declared that no competing interests exist.

the right dosage of prescribed morphine for a child who consistently reported of moderate to severe pain.

## Conclusion

Final year nursing students and nurses have unsatisfactory knowledge and attitudes regarding children's pain; which reiterates the need for urgent and effective educational efforts in this area. Regular in-service training should be offered to post-registration nurses to enhance their pediatric pain knowledge and attitudes for improved pain care in children.

## Introduction

Children have a right to adequate pain assessment and management by qualified healthcare professionals including nurses [1]. Optimal pain relief ought to be a priority in healthcare especially for children, as they may be developmentally challenged in communicating about their pain [2]. In spite of this, children's pain has been reported in the international literature as regularly under-assessed and under-treated [3–5]. In sub-Saharan Africa, children have to cope with abundance of pain-associated diseases such as sickle cell disease [6], suboptimal healthcare systems and poverty [7,8]; all of which worsen the plight of vulnerable children.

In Ghana, nursing students go through three or four years of training for a diploma or Bachelor's degree respectively. Afterwards, they are required to not only pass their licensing examination but also undertake a one-year clinical rotation before qualifying as registered nurses [9]. During the clinical rotation period, newly qualified nurses are expected to work and enhance their nursing competencies in varied clinical departments including pediatric care settings. In the course of their clinical rotation, they will come into contact with children in pain, and their families, and will have to rely on their knowledge and demonstrate appropriate attitudes to offer the needed support.

Specialization in children's nursing is not required of nurses to practice within pediatric care settings in Ghana. Many clinical nurses encountered in children's care units are general nurses as the pediatric nursing specialty is still new and evolving in Ghana. Nurses who work in the children's units of hospitals are also confronted with children who have varied levels of pain caused by clinical conditions, injuries, hospital procedures or some unknown factors. Conversely, children and their families equally have higher expectations of nurses and other healthcare professionals to be able to recognize, assess and manage pain in their daily work activities [10]. Thus, knowledge and attitude of nurses relating to children's pain assessment and management is very essential. In light of this, the International Association for the Study of Pain advocates for the best possible pain education during professional training and post-qualification [11].

Knowledge and attitudes of nursing students and nurses have been examined in previous literature. Nursing students' knowledge and attitudes regarding children's pain was found to be inadequate in studies that were conducted in Mexico [12] and Egypt [13] using the Pediatric Nurses' Knowledge and Attitudes Regarding Pain (PNKAS) instrument. Unsatisfactory knowledge and attitudes of nurses have also been reported using the PNKAS tool in countries including the United States of America [14,15], Turkey [16], Mexico [12], Norway [17] and Saudi Arabia [18]. It seems from the review of earlier research studies that, nursing students and nurses' knowledge and attitudes on children's pain have not been examined from the sub-Saharan African region. Moreover, the results from previous studies indicate varied areas of

deficiencies in children's pain knowledge and attitude which may not reflect those of nursing students and nurses within the sub-Saharan African region.

The current study therefore assessed and compared nursing students and nurses' knowledge and attitudes pertaining to children's pain in the Ghanaian context. The goal of this was to have a baseline information to guide the development and implementation of a sustainable short-course programme for nursing students and nurses in Ghana.

## Materials and methods

### Study design, setting and participants

A comparative cross-sectional survey was conducted among final year nursing students and nurses in Ghana. Four nursing educational institutions and eight hospitals were purposively chosen as study sites. Consideration was given to type of ownership and the geographical location of these educational and hospital facilities.

The nursing educational institutions comprised of two universities (one private and one public) and two nursing training colleges (NTCs) (one private and one public). These educational institutions are accredited by the National Accreditation Board (NAB) and the National Council for Tertiary Education (NCTE) of Ghana to run undergraduate nursing programs leading to the award of Diploma (in the case of the NTC trainees) or Bachelor's degree (in the case of the university trainees). A review of the training curricula for the nursing students revealed that pain management is part of their general nursing and practical courses but not situated in the context of most of the courses dedicated to children's nursing.

The eight hospitals comprised of three specialist children's hospitals, two district hospitals, a regional hospital, a university (quasi-government) hospital and a mission (catholic) hospital. All hospitals had both an out-patient department and in-patient facility where they cared for hospitalized children and their families. Nurses are expected to assess and manage children's pain as part of their roles during vital signs monitoring and routine care activities in these hospitals. As at the time of the study, the number of registered nurses at the pediatric units of the eight facilities was 70. These nurses were being assisted by rotation nurses, student nurses and healthcare assistants to discharge their duties. Rotation nurses are interns who have passed their licensing examinations and undergoing post-qualification training in specified clinical areas before receiving their registration.

The appropriate sample size for the study was guided by Yamane's [19] formula for cross-sectional studies assuming a 95% confidence interval and a type I error rate of 5%. Based on an estimated population of 1100 final year nursing students and 70 nurses, a minimum of 293 nursing students and 60 nurses were considered appropriate in powering the study to draw conclusions with a higher level of precision. Nine hundred (900) students who were in the first semester of their final year nursing program and 70 nurses were purposefully approached, of which 554 nursing students and 65 nurses respectively agreed and participated in the study, giving an overall response rate of 63.8% (61.6% for the nursing students and 92.9% for the nurses).

### Data collection procedures, instrument and analysis

The researchers approached nursing students and nurses at their recess times between lectures and nursing duties respectively to brief them about the study's objective and characteristics. Those who consented to participate were given hard copies of the questionnaire which took about 20–30 minutes to complete. Over the course of three months, demographic and primary data on participants' pediatric pain knowledge and attitudes were collected using the Pediatric Nurses Knowledge and Attitude Survey Regarding Pain (PNKAS) instrument [20].

The PNKAS instrument consists of 42 questions: 25 binary (true or false) response type questions, 13 multiple choice questions (MCQs) and two case studies expanded into four questions. Responses given by the participants in the survey were scored one (1) for a correctly answered knowledge question and/ a positive attitude or both; and zero (0) for a wrongly answered question and/ a negative attitude. Thus, the possible attainable scores on the PNKAS instrument ranged from zero to 42. Individually attained scores were converted into percentages for standardization purposes using the formula: (obtained scores/ 42) * 100. In line with earlier studies [21–23], a total score of 80% was considered as a satisfactory level of knowledge and attitudes regarding children's pain.

Validity and reliability of the PNKAS tool has been reported in Manworren's [13] study. Face and content validity of the tool was established by a panel of five nursing experts in pain management. Test-retest reliability among 6 nurses and 6 child life specialists was reported at a correlation coefficient of 0.67, indicating an appreciable level of instrument stability. Cronbach's alpha of the instrument using data from two distinct groups of pediatric nurses ranged from 0.72 to 0.77, signifying an acceptable level of internal consistency. In the current study, face validity of the tool was assessed by 11 pediatric experts (six registered nurses, four nurse educators and one pediatrician) who recommended its usage in Ghana.

Participants' data were initially entered and cleaned in Microsoft excel before being exported into Statistical Package for Social Sciences version 25 for further analysis. The ages of the final year nursing students were categorised into two: regular (up to 24 years) and matured (those aged 25 years and above). The ages of the nurses were also classified into two: those up to 30 years and those above 30 years. Categorical variables were presented using frequencies and percentages, while continuous variables were reported with means, standard deviations and ranges. Chi-square test of independence analysis was used to examine the association between nursing status (nursing students and nurses) and gender (male and female). Differences in PNKAS scores between any two groups were examined using the independent samples t-test. Differences in PNKAS scores between three or more groups was analysed using the one-way analysis of variance (ANOVA) method. Statistical significance for all test was set at the .05 level.

## Ethical considerations

Ethical approval with reference number CHRPE/AP/574/18 was obtained from the Committee on Human Research, Publication and Ethics. Administrative approvals were also obtained from the respective hospitals and educational institutions involved in the study. Class representatives and ward-in-charges were consulted for permission to engage and distribute the questionnaires during each data collection session. Anonymity, confidentiality and participants' right to voluntary participation were ensured during data collection.

## Results

### Participants' socio-demographic characteristics

The mean age of the nursing students was 24 years with a standard deviation of 4 years; their ages ranged from 21 to 47 years. Slightly over three-quarters of the nursing students were females (n = 418, 75.5%) while males (n = 136) formed 24.5%. A greater proportion of the nursing students were enrolled in the Diploma program (n = 355, 64.1%), whereas those in the Bachelor's degree program (n = 199) comprised of 35.9%.

On the other hand, the mean (SD) age of the nurses was 29 (4) years with a range of 23 to 38 years. Over four-fifths of the nurses were females (n = 53, 81.5%) while the males (n = 12) formed 18.5%. Majority of the nurses had a diploma as their highest education qualification

(n = 42, 64.6%); this was followed by Certificate holders (n = 13, 20%) with only 15.4% having attained a Bachelor's or Master's degree (n = 10). A greater number of the nurses had worked for up to five years in the profession (n = 41, 63.1%), whereas 36.9% of them had worked for more than five years (n = 24). Almost two-thirds of the nurses were working in general hospitals (n = 43, 66.2%) whereas the remaining one-third were working in specialist children's hospitals (n = 22, 33.8%). A cross-tabulation analysis revealed that the nurses who were working in general hospitals had relatively higher educational qualification than those who were working in the specialist children's hospital (18.6% versus 9.1% had obtained their Bachelor's and Master's degrees).

There were no statistically significant differences in the gender distribution of both nursing students and nurses (p = .276). However, nursing students were significantly younger than the nurses (p < .001).

## Nursing students' knowledge and attitudes regarding children's pain

On the 42-item PNKAS instrument, an average (SD) correct answer score of 42.1% (8.0%) was obtained by nursing students, with a minimum correct score of 21.4% and a maximum of 81.0%. The five questions most often answered correctly by nursing students were focused on the subjective and multi-dimensional nature of the pain experience and its treatment, the importance of pre-emptive analgesia and nonpharmacological pain management intervention (Table 1). The top five questions most often answered incorrectly by the nursing students were concentrated on cancer-related pain medications, opioid drug administration, pain assessment and pain perception (Table 2).

Results of an independent samples t-test analyses revealed that, there were no statistically significant differences in the mean PNKAS scores on the basis of the nursing students' age classification (p = .311) and gender (male and female) (p = .158). However, there was a statistically significant difference in the mean PNKAS scores of the nursing students on the basis of their

**Table 1. Items most often answered correctly by participants (n = 619).**

| Items *(correct answer)* | Students (n = 554) | Nurses (n = 65) |
|---|---|---|
| | f (%) Correct; Rank | f (%) Correct; Rank |
| Comparable stimuli in different people produce the same intensity of pain *(F)* | 434 (78.3); 1st | 41 (63.1); 5th |
| Children who will require repeated painful procedures should receive maximum treatment for the pain and anxiety of the first procedure to minimize the development of anticipatory anxiety before subsequent procedures *(T)* | 433 (78.2); 2nd | 45 (69.2); 3rd |
| Combining analgesics and non-drug therapies that work by different mechanisms may result in better pain control with fewer side effects than using a single analgesic agent *(T)* | 386 (69.7); 3rd | 48 (73.8): 1st |
| After the initial recommended dose of opioid analgesic, subsequent doses should be adjusted in accordance with the individual patient's response *(T)* | 384 (69.3); 4th | 47 (72.3); 2nd |
| Parents should not be present during painful procedures *(F)* | 374 (67.5); 5th | – |
| The child/adolescent with pain should be encouraged to endure as much pain as possible before resorting to an opioid for pain *(F)* | – | 43 (66.2); 4th |

Note: f–frequency; %–percentage; T–True; F–False

**Table 2. Items most often answered incorrectly by participants (n = 619).**

| Items *(correct answer)* | Students (n = 554) | Nurses (n = 65) |
|---|---|---|
| | f (%) Incorrect; Rank | f (%) Incorrect; Rank |
| Which of the following drugs are potentially useful for treatment of children's cancer pain? *(All of the following: Ibuprofen, Hydromorphone, Amitriptyline)* | 549 (99.1); 1st | |
| The likelihood of opioid addiction occurring due to pain treatment with opioid analgesics is *(<1%)* | 516 (94.9); 2nd | |
| Two hours after a child received morphine 2 mg IV, his pain ratings consistently ranged from 6 to 8 with no clinically significant side effects. His physician's order for analgesia is "morphine IV 1–3 mg q1h PRN pain relief". The most appropriate action by the nurses is to: *(Administer morphine 3 mg IV now)* | 511 (92.2); 3rd | 65 (100.0); 1st |
| A 15-year old Andrew smiles on his first day post-abdominal operation. Upon entering his room, he smiles with you and continues talking and joking with his visitor. He rates his pain as 8 on a scale of 0 to 10 (0 = no pain, 10 = worst pain) during assessment, how would you rate Andrew's pain? *(8)* | 473 (85.4); 4th | 60 (92.3); 5th |
| The percentage of patients who over report the amount of pain they have is: *(0–10%)* | 456 (82.3); 5th | 62 (95.4); 4th |
| The usual time to peak effects for traditional analgesics given orally is: *(30 minutes)* | | 65 (100.0); 1st |
| Which of the following IV doses of morphine administered would be equivalent to 15 mg of oral morphine? *(Morphine 5 mg IV)* | | 65 (100.0); 1st |

Note: f–frequency; %–percentage; mg–milligram; IV–Intravenous; q1h –Hourly; PRN–When necessary

study level (p < .001). Nursing students who were enrolled on the Bachelor's degree program (M = 43.9%, SD = 8.9%) had significantly higher PNKAS scores than their counterparts who were pursuing the Diploma program (M = 41.1%, SD = 7.3%).

## Nurses' knowledge and attitudes regarding children's pain

Nurses had a mean (SD) correct answer score of 36.7% (6.9%) on the 42-item PNKAS scale, with a minimum correct score of 21.4% and a maximum of 57.1%. The five questions most often answered correctly by the surveyed nurses were related to the subjective and multi-dimensional nature of the pain experience and its treatment as well as the benefits of pre-emptive analgesia (Table 1). The top five (5) items where nurses frequently answered incorrectly were centered on pharmacokinetics, pain assessment and pain perception (Table 2).

Results of an independent samples t-test and one-way analysis of variance (ANOVA) showed that, there were no statistically significant differences in the mean PNKAS scores on the basis of the nurses' age classification (p = .164), gender (male and female) (p = .674), highest educational qualification (certificate, diploma, Bachelor's or Master's degree) (p = .798) and working years in the nursing profession (up to five years and those above five years) (p = .638). Nevertheless, there was a statistically significant difference in the mean PNKAS scores of the nurses on the basis of the type of hospital they were working in (p < .001). Nurses who were working in general hospitals (M = 38.8%, SD = 7.0%) had significantly greater PNKAS scores than those who were working in specialist pediatric hospitals (M = 32.7%, SD = 4.5%).

## Comparison of nursing students and nurses' pain knowledge and attitudes

Results of an independent samples t-test revealed that, nursing students [M = 17.69 (42.1%), SD = 3.37 (8.0%)] correctly scored higher on the PNKAS instrument than registered nurses [M = 15.43 (36.7%), SD = 2.88 (6.9%)], t (617) = 5.18, p < .001, two-tailed. The difference of 2.26 (5.4%) scale units indicated a small but statistically significant effect (scale range: 0 to 42) and the 95% confidence interval around the difference between the group means was relatively precise [1.40 (3.3%) to 3.12 (7.4%)].

There were statistically significant differences between nursing students and nurses in 13 items on the PNKAS instrument (Table 3). Nursing students significantly scored higher than

**Table 3. Comparison of nursing students and nurses' knowledge and attitudes regarding children's pain.**

| Items *(correct answer)* | Student (n = 554) f (%) Correct | Nurse (n = 65) f (%) Correct | p value |
|---|---|---|---|
| Comparable stimuli in different people produce the same pain intensity *(F)* | 434 (78.3) | 41 (63.1) | .017 |
| Sedation always precedes opioid related respiratory distress *(F)* | 217 (39.2) | 35 (53.8) | .023 |
| The recommended route of administration of opioid analgesics to children with prolonged cancer-related pain is: *(Intravenous)* | 238 (43.0) | 12 (18.5) | < .001 |
| The recommended route of administration of opioid analgesics to children with brief, severe pain of sudden onset is: *(Intravenous)* | 228 (41.2) | 18 (27.7) | .027 |
| Analgesics for post-operative pain should initially be given: *(Around the clock on a fixed schedule)* | 217 (39.2) | 6 (9.2) | < .001 |
| Analgesia for chronic cancer pain should be given: *(Around the clock on a fixed schedule)* | 285 (51.4) | 10 (15.4) | < .001 |
| The most likely explanation for why a child/adolescent with pain would request increased pain medication doses is that: *(He/ she is experiencing increased pain)* | 268 (48.4) | 16 (24.6) | < .001 |
| Which of the following drugs are useful for treating cancer pain? *(All of the above)* | 5 (0.9) | 13 (20.0) | < .001 |
| Which of the following describes the best approach for cultural considerations in caring for child/ adolescent in pain? *(Children/ adolescents should be individually assessed to determine cultural influences on pain)* | 209 (37.7) | 36 (55.4) | .006 |
| What do you think is the percentage of patients who over report the amount of pain they have? *(0%)* | 98 (17.7) | 3 (4.6) | < .001 |
| The usual time to peak effects for analgesics given orally is: *(30 minutes)* | 294 (53.1) | 0 (0.0) | < .001 |
| Which of the following intravenous (IV) doses of morphine administered would be equivalent to 15 mg of oral morphine? *(Morphine 5 mg IV)* | 135 (24.4) | 0 (0.0) | < .001 |
| Two hours after a child received morphine 2 mg IV, his pain ratings consistently ranged from 6 to 8 and he had no clinically significant side effects. His physician's order for analgesia is "morphine IV 1–3 mg q1h PRN pain relief." The most appropriate action by the nurses is to: *(Administer morphine 3 mg IV now)* | 210 (37.9) | 0 (0.0) | < .001 |

Note: f–Frequency; %–Percentage; p–the degree of making a type I error; T–True; F–False

nurses in 10 out of the 13 identified areas ($p < .05$). These were noted in items that addressed pain physiology, pharmacokinetics, pharmacology of analgesics and pain perceptions. On the other hand, nurses had greater scores in 3 out of the 13 identified items on the PNKAS instrument ($p < .05$). These were observed in the following items: the relationship between sedation and opioid-related respiratory distress, useful cancer-related pain medications and cultural influences on children's pain.

## Discussion

With the goal of developing and implementing the content for a sustainable educational programme (short-course) on pediatric pain assessment and management, the present study sought to assess and compare nursing students' and nurses' knowledge and attitudes regarding children's pain in the Ghanaian context. Our findings revealed that, nursing students and nurses generally have insufficient knowledge and attitudes towards pain management in children relative to the 80% cut off mark [21–23]. There appears to be an internationally recognized knowledge and attitudes deficits among nursing students and nurses on this subject [12–18]. One of the possible reasons for this trend may be related to the insufficient pediatric pain management training in nursing curricula [24,25]. Limited continuing educational opportunities for post-registration nurses on this subject may also be responsible for these outcomes [18,26,27]. Due to these insufficiencies in knowledge and attitudes, pain in children globally remains inadequately managed, which leads to unnecessary suffering in this vulnerable population.

Nursing students who were enrolled on the Bachelor's degree program had significantly higher pediatric pain knowledge and attitudes than their counterparts who were pursuing the Diploma program. This is similar to the findings of Plaisance and Logan [28] whereby nursing students in the baccalaureate degree program had a significantly higher pain knowledge and attitudes scores compared to those in the associate degree programs. Possibly, the content and duration of training may play a role in the final year nursing students' acquisition of knowledge and attitudes on pediatric pain in the current study. On the other hand, nurses who were working in general hospitals also had higher pediatric pain knowledge and attitudes than those who were working in specialist pediatric hospitals. This may be attributed to the fact that a greater percentage of the nurses who worked in the general hospitals had relatively higher educational background than those in the specialist pediatric hospitals. This is not surprising as the level of education has been identified as one of the influencing factors in pain knowledge and attitudes [29,30].

In contrast to Ortiz et al.'s [12] findings, nursing students in the current study significantly had higher scores than the nurses on the entire PNKAS instrument and also in 10 individual items out of the 13 identified areas of significant differences. The relatively recent knowledge acquisition by the nursing students may be responsible for these findings. The lack of regular in-service educational courses on pediatric pain management post-qualification may also account for the observed trend [31].

Nursing students and nurses had similarities in their top and bottom five pediatric pain knowledge and attitudes. Consistent with earlier literature [13,17], majority of the sampled nurses and nursing students had good pediatric pain knowledge and attitudes in items that measured the benefits of pre-emptive analgesia as well as the subjective and multi-dimensional nature of the pain experience and its treatment. A greater proportion of participants in both groups performed poorly on items that were related to opioid drug administration, pain assessment and pain perception, supporting previous research findings [12,17,18]. These identified areas of sufficient and insufficient knowledge and attitudes should be considered during the design and implementation of educational programs on this subject.

Differences were also observed in the top and bottom five pediatric pain knowledge and attitudes among the two groups. Relative to the nurses, many of the nursing students recognized the relevance of parental support as an important advocacy strategy for addressing the pain concerns of children [32] during painful procedures, which is consistent with the findings of Gadallah and colleagues [13]. Similar to earlier literature [16–18,33], almost all the nursing students could not identify useful cancer-related pain medications and overestimated the likelihood of narcotic addiction in patients who are being treated with opioids. These findings are not surprising as nursing students in Ghana have limited care experiences with children with cancers and appear to harbour misconceptions on narcotics. Recent evidence however, points to a well-tolerated and effective opioid use among children in practice [34,35]. Compared with the nursing students, a greater proportion of the nurses acknowledged the importance of early pain treatment in children as advocated for by the International Association for the Study of Pain (IASP) as a fundamental human right [1].

It is significant to also note that, all the surveyed nurses could not accurately determine the onset of action of orally administered analgesics, equianalgesic dose of orally administered morphine, and the right dosage of prescribed morphine for a child who consistently reported of moderate to severe pain. These results are of great concern as these nurses are already practising in hospitals and may end up mismanaging children's pain. As the problem of poor assessment and management of children's pain persists among nurses, there is an urgent need to develop, implement and evaluate pain educational interventions especially in resource constrained setting to improve pain management for vulnerable children and their families.

## Strengths and limitations of the study

To the best of our knowledge on existing literature, this study is the first of its kind in Ghana and sub-Saharan Africa. The adequate sample size that was used in the current study reassures that the study was sufficiently powered to support the findings, which can be applied in other similar settings. The inclusion of diverse educational institutions and healthcare facilities also adds to the reliability of the studied outcomes.

One of the drawbacks of the current investigation was that, participants were purposively chosen based on their availability at the research sites without being randomly selected. The surveyed nurses were relatively younger in age with shorter years of experience than the general nursing population; this should be considered when interpreting the study's findings. In spite of these limitations, the present study provides useful information on nursing students and nurses' knowledge and attitudes pertaining to children's pain management in the Ghanaian context.

## Implications

The study has implications for nursing education, practice, policy, and research. Pediatric pain assessment and management in nursing curricula needs to be strengthened further. Educational approaches that seek to reinforce positive knowledge and attitude translation in practice should also be applied. Such programs should explore areas of higher and lower educational needs to target and reinforce outcomes. Nurses should be encouraged to engage in self-directed learning activities and continual professional educational opportunities to improve pediatric pain management.

Existing guidelines on pain management in children should also be enforced to guide nursing practice. Future studies should use qualitative research approaches to gain an understanding on the factors that underpin nursing knowledge and attitudes on the subject.

Observational studies on how pain is assessed and managed in hospitalized children would also be insightful and assist in guiding future educational interventions.

## Conclusion

Final year nursing students and nurses have unsatisfactory knowledge and attitudes regarding children's pain; which reiterates the need for urgent and effective educational efforts in this area. Regular in-service training should be offered to post-registration nurses to enhance their pediatric pain knowledge and attitudes for improved pain care in children.

## Supporting information

**S1 Table. This is the S1 Table codebook for nursing students and nurses' knowledge and attitude regarding children's pain: A comparative cross-sectional study in Ghana.** (XLSX)

## Acknowledgments

Our profound appreciation goes to the management of the educational institutions and health facilities for their co-operation. We are also grateful to the nursing students and nurses who took time off their very busy schedules and participated in the study.

## Author Contributions

**Conceptualization:** Abigail Kusi Amponsah, Victoria Bam, Joana Kyei-Dompim, Anna Axelin.

**Data curation:** Abigail Kusi Amponsah, Evans Oduro, Joana Kyei-Dompim, Collins Kwadwo Ahoto.

**Formal analysis:** Abigail Kusi Amponsah, Evans Oduro, Victoria Bam, Joana Kyei-Dompim, Collins Kwadwo Ahoto, Anna Axelin.

**Funding acquisition:** Abigail Kusi Amponsah, Victoria Bam.

**Investigation:** Abigail Kusi Amponsah, Evans Oduro, Victoria Bam, Joana Kyei-Dompim, Collins Kwadwo Ahoto, Anna Axelin.

**Methodology:** Abigail Kusi Amponsah, Evans Oduro, Victoria Bam, Joana Kyei-Dompim, Collins Kwadwo Ahoto, Anna Axelin.

**Project administration:** Abigail Kusi Amponsah, Anna Axelin.

**Resources:** Abigail Kusi Amponsah, Victoria Bam, Anna Axelin.

**Software:** Abigail Kusi Amponsah.

**Supervision:** Victoria Bam, Anna Axelin.

**Validation:** Abigail Kusi Amponsah, Evans Oduro, Victoria Bam, Joana Kyei-Dompim, Collins Kwadwo Ahoto, Anna Axelin.

**Visualization:** Abigail Kusi Amponsah, Evans Oduro, Victoria Bam, Joana Kyei-Dompim, Collins Kwadwo Ahoto, Anna Axelin.

**Writing – original draft:** Abigail Kusi Amponsah, Evans Oduro, Joana Kyei-Dompim.

**Writing – review & editing:** Abigail Kusi Amponsah, Evans Oduro, Victoria Bam, Joana Kyei-Dompim, Collins Kwadwo Ahoto, Anna Axelin.

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
