## [Decision Letter · Decision Letter 0]

12 Aug 2019

PONE-D-19-18155

Nursing Students and Nurses’ Knowledge and Attitudes regarding Children’s Pain: A Comparative Cross-sectional study

PLOS ONE

Dear

   Abigail Kusi Amponsah 

,

Thank you for submitting your manuscript to PLOS ONE. After careful consideration, we feel that it has merit but does not fully meet PLOS ONE’s publication criteria as it currently stands. Therefore, we invite you to submit a revised version of the manuscript that addresses the points raised during the review process.

We would appreciate receiving your revised manuscript by 15th September. To enhance the reproducibility of your results, we recommend that if applicable you deposit your laboratory protocols in protocols.io, where a protocol can be assigned its own identifier (DOI) such that it can be cited independently in the future. For instructions see: http://journals.plos.org/plosone/s/submission-guidelines#loc-laboratory-protocols

We look forward to receiving your revised manuscript.

Kind regards,

Sharon Mary Brownie

Academic Editor

PLOS ONE

Journal Requirements:

3. Please provide additional details regarding participant consent. In the ethics statement in the Methods and online submission information, please ensure that you have specified (1) whether consent was suitably informed and (2) what type you obtained (for instance, written or verbal). If your study included minors under age 18, state whether you obtained consent from parents or guardians. If the need for consent was waived by the ethics committee, please include this information.

4. Please include in your Methods section the date ranges over which you recruited participants to this study.

Additional Editor Comments:

Please pay careful attention to the detailed feedback provided by the reviewers along with the recommendation to simplify some of the tables.

Reviewers' comments:

Reviewer's Responses to Questions

**Comments to the Author**

1. Is the manuscript technically sound, and do the data support the conclusions?

Reviewer #1: Yes

Reviewer #2: Yes

2. Has the statistical analysis been performed appropriately and rigorously? 

Reviewer #1: No

Reviewer #2: I Don't Know

3. Have the authors made all data underlying the findings in their manuscript fully available?

Reviewer #1: Yes

Reviewer #2: Yes

4. Is the manuscript presented in an intelligible fashion and written in standard English?

Reviewer #1: Yes

Reviewer #2: Yes

5. Review Comments to the Author

Reviewer #1: Overall Comments

Thank you for the opportunity to review the manuscript. The manuscript is well written and addresses a neglected area in nursing care. It provides interesting evidence that when used could help improving the paediatric pain management by nurses. Being among the first studies in Africa, it also forms a basis for future research on pain management in the continent and globally.

Despite its strengths, the study has significant methodological issues. These include (i) lack of a clear operational definition of level of knowledge and attitude and (ii) failure to assess the effects of potential confounders on the level of knowledge and attitudes. The first methodological flaws has the potential to invalidate the findings reported while the second one limits the interpretation of the findings and potential use of the findings to inform necessary interventions especially in the short-term which is was the aim of the authors for conducting this study.

Specific Comments

The abstract according to the journal should be structured (with subheadings – introduction, methods, results, conclusion).

Line 31: The authors should be specific of the statistical analyses performed.

Line 57: “In Ghana, nursing students….” Kindly clarify the level of study (diploma or degree or both)

Lines 81-83: Consider revising to sub-Saharan Africa rather than Africa since Egypt is in Africa. Also, revise the sentence to reflect the correct position/ state. Could studies on nursing students and nurses’ knowledge and attitude on children pain management have been carried out somewhere in Africa but the studies cannot be accessed by the authors?

Line 113: Review to ensure that the number of nurses (70) is correct. Are all the nurses in the eight sampled health facilities 70 in number?

Line 117: How did the author calculate the response rate? Was it based on the target population of 1100 and 70 or based on the sample size (which should be the case)? Revise or clarify on the response rate.

Under study participants’, the author should highlight the study semester for the final year nursing students. Was it the last semester of the final year or was it at the start of the final year?

Line 134-135: “Higher scores on PNKAS instrument denotes a satisfactory level of knowledge and attitudes….” The statement is unclear and non-specific. The authors should operationalise levels of knowledge and attitude. E.g. What are the cut-off points? What satisfactory and unsatisfactory levels of knowledge and attitude?

The study is a comparative study; however, the authors only focus on comparing nursing students and nurses. The authors do not assess the effects of potential confounders on the level of knowledge and attitudes such as age or years of practice, gender, study level (diploma vs degree), clinical area of practice (paediatric, medical, orthopaedics etc.) and type of hospitals (private vs public) for nurses; and gender, study level (diploma vs degree), training institutions for nursing students. Further analyses should be performed to reflect the potential effects of these confounders. The findings of these additional analyses are key especially if the results of this study are to be used as a baseline for developing educational programmes for nurses and nursing students. Overall finding might be masking some of the effects of potential confounders e.g. the authors in lines 206-207 report that a small difference of 2.26 was found to be statistically significant; however, they do not perform additional tests to assess what would be contributing to these small difference being statistically significant.

Table 1 and 2 should be dropped since all the information on the two tables has been described in text.

Lines 164, 178, 187, 190, 197, 200, 201, 211: Remove the word see or refer in the “see table” or “refer to table..”

Line 26 and 221: The authors should specify what kind of a nursing education programme the study aims to develop with the information. Is the program specific to pain? Is it a short-term or long-term program? Are the findings of these study alone sufficient to be used for developing a nursing education program?

Line 237-239: “….beyond the scope of the current study.” The information referred to (level of education, years of experience among others) would be basic demographic information for a study among nurses. Based on the data available, the authors should consider performing additional analyses as recommended above. Otherwise, this should be stated as a major limitation of the study because it would affect the overall interpretation of the results.

In most jurisdiction, continuous professional development (CPD) is a prerequisite for renewal of practicing licence for health professionals. While, the CPDs may not be specific to children pain management, the authors should highlight how CPDs or in-service trainings are incorporated in the health care system in Ghana in discussion line 233-241.

Line 301-302: The implication stated is not supported by the study findings and should be deleted. However, the next statement on implication (Line 302-303) indicates that there are policies already in place and that what is needed is strengthening their implementation.

Line 313-317: The sentences do not fit into the conclusion since they are repetitions of the discussion. The conclusion used in the abstract best summarises the findings of the study and concludes the study. Revise to include aspect from the abstract in the main text conclusion.

Reviewer #2: The manuscript is written in clear and correct English. I presents original research which is valuable and will add to the body of nursing knowledge in regard to pain assessment and mangement

6. PLOS authors have the option to publish the peer review history of their article (what does this mean?). If published, this will include your full peer review and any attached files.

Reviewer #1: No

Reviewer #2: No

---

## [Author Response · Author response to Decision Letter 0]

16 Sep 2019

Reviewer #1: Overall Comments

Thank you for the opportunity to review the manuscript. The manuscript is well written and addresses a neglected area in nursing care. It provides interesting evidence that when used could help improving the paediatric pain management by nurses. Being among the first studies in Africa, it also forms a basis for future research on pain management in the continent and globally.

Despite its strengths, the study has significant methodological issues. These include (i) lack of a clear operational definition of level of knowledge and attitude and (ii) failure to assess the effects of potential confounders on the level of knowledge and attitudes. The first methodological flaws has the potential to invalidate the findings reported while the second one limits the interpretation of the findings and potential use of the findings to inform necessary interventions especially in the short-term which is was the aim of the authors for conducting this study.

Response

Thank you for the feedback. The issues raised have been revised in the manuscript on page 7, lines 147-149; page 10, lines 212-218 and page 13, lines 231-239.

Specific Comments

Comment

The abstract according to the journal should be structured (with subheadings – introduction, methods, results, conclusion).

Response

The abstract has been organised using the suggested subheadings. Refer to page 2 lines 23, 30, 37 and page 3 line 47.

Comment

Line 31: The authors should be specific of the statistical analyses performed.

Response

As suggested, this has been amended on page 2, lines 34-35.

Comment

Line 57: “In Ghana, nursing students….” Kindly clarify the level of study (diploma or degree or both)

Response

This has been clarified on page 3, lines 62-63.

Comment

Lines 81-83: Consider revising to sub-Saharan Africa rather than Africa since Egypt is in Africa. Also, revise the sentence to reflect the correct position/ state. Could studies on nursing students and nurses’ knowledge and attitude on children pain management have been carried out somewhere in Africa but the studies cannot be accessed by the authors?

Response

As suggested, this has been revised and can be found on page 4, line 88; page 5 lines 90-91. 

To the best of our review, we did not find any additional published studies other than the ones indicated. Access was not a barrier to study identification due to the wealth of databases provided by the University of Turku Library. 

Comment

Line 113: Review to ensure that the number of nurses (70) is correct. Are all the nurses in the eight sampled health facilities 70 in number?

Response

As at the time of the study, the number of registered nurses at the paediatric units of the eight facilities were 70. These nurses were being assisted by rotation nurses, student nurses and healthcare assistants. Rotation nurses are interns who have passed their licensing results and undergoing post-qualification training in specified clinical areas before receiving their registration. The above clarification has been provided on page 6, lines 118-123.

Comment

Line 117: How did the author calculate the response rate? Was it based on the target population of 1100 and 70 or based on the sample size (which should be the case)? Revise or clarify on the response rate.

Response

Thank you for your comments. Per our reference [1], the response rate is calculated based on the number of participants who are approached and not on the entire study population nor the calculated sample size. We have therefore maintained this as presented on page 6, line 129-132 as we approached 900 nursing students and all 70 nurses. 

Reference

1. Phillips AW, Friedman BT, Durning SJ. How to Calculate a Survey Response Rate. Acad Med. 2017;92: 269. doi:10.1097/ACM.0000000000001410

Comment

Under study participants’, the author should highlight the study semester for the final year nursing students. Was it the last semester of the final year or was it at the start of the final year?

Response

As suggested, this has been revised on page 6, lines 129-130.

Comment

Line 134-135: “Higher scores on PNKAS instrument denotes a satisfactory level of knowledge and attitudes….” The statement is unclear and non-specific. The authors should operationalise levels of knowledge and attitude. E.g. What are the cut-off points? What satisfactory and unsatisfactory levels of knowledge and attitude?

Response

As suggested, this has been revised on page 7, lines 147-149.

Comment

The study is a comparative study; however, the authors only focus on comparing nursing students and nurses. The authors do not assess the effects of potential confounders on the level of knowledge and attitudes such as age or years of practice, gender, study level (diploma vs degree), clinical area of practice (paediatric, medical, orthopaedics etc.) and type of hospitals (private vs public) for nurses; and gender, study level (diploma vs degree), training institutions for nursing students. Further analyses should be performed to reflect the potential effects of these confounders. The findings of these additional analyses are key especially if the results of this study are to be used as a baseline for developing educational programmes for nurses and nursing students. Overall finding might be masking some of the effects of potential confounders e.g. the authors in lines 206-207 report that a small difference of 2.26 was found to be statistically significant; however, they do not perform additional tests to assess what would be contributing to these small difference being statistically significant.

Response

As suggested, further analysis have been performed and reported on page 8, lines 165-170; page 9, lines 183-185, lines 189-197; page 10, lines 212-218; page 13 lines 231-239.

Comment

Table 1 and 2 should be dropped since all the information on the two tables has been described in text.

Response

As suggested, Table 1and 2 have been removed.

Comment

Lines 164, 178, 187, 190, 197, 200, 201, 211: Remove the word see or refer in the “see table” or “refer to table..”

Response

As suggested, these have been removed. Refer to page 10, lines 207, 210; page 11, line 219; page 12, line 221; page 13, lines 228-229; page 14 line 254; page 15 line 261.

Comment

Line 26 and 221: The authors should specify what kind of a nursing education programme the study aims to develop with the information. Is the program specific to pain? Is it a short-term or long-term program? Are the findings of these study alone sufficient to be used for developing a nursing education program?

Response

This has been clarified on page 2, lines 27-28; and page 17, lines 264-265.

Comment

Line 237-239: “….beyond the scope of the current study.” The information referred to (level of education, years of experience among others) would be basic demographic information for a study among nurses. Based on the data available, the authors should consider performing additional analyses as recommended above. Otherwise, this should be stated as a major limitation of the study because it would affect the overall interpretation of the results.

Response

As suggested, further analysis has been performed and reported on page 8, lines 165-170; page 9, lines 183-185, lines 189-197; page 10, lines 212-218; page 13 lines 231-239.

Comment

In most jurisdiction, continuous professional development (CPD) is a prerequisite for renewal of practicing licence for health professionals. While, the CPDs may not be specific to children pain management, the authors should highlight how CPDs or in-service trainings are incorporated in the health care system in Ghana in discussion line 233-241.

Response

As suggested, this has been clarified on page 18, line 294.

Comment

Line 301-302: The implication stated is not supported by the study findings and should be deleted. However, the next statement on implication (Line 302-303) indicates that there are policies already in place and that what is needed is strengthening their implementation.

Response

As suggested, this has been removed; refer to pages 20-21.

Comment

Line 313-317: The sentences do not fit into the conclusion since they are repetitions of the discussion. The conclusion used in the abstract best summarises the findings of the study and concludes the study. Revise to include aspect from the abstract in the main text conclusion.

Response

As suggested, this has been amended on page 21, lines 362-365.

Comment

Reviewer #2: The manuscript is written in clear and correct English. I presents original research which is valuable and will add to the body of nursing knowledge in regard to pain assessment and management.

Response

Thanks for the feedback.

Reviewer’s comments 

Comment

Thank you for the opportunity to review the manuscript.

Response

Thanks for the feedback.

Comment

The manuscript presents original work which is presented in correct language, the methodology and results are in congruence. This piece of work will add to the knowledge in regards to management of pain in children in Ghana.

Response

Thanks for the feedback.

Comment

Tables 1-4 discussions are quite lengthy. There is need to refine the results’ discussion of tables 1-4 to reduce wording, so as to make it easy for the reader.

Response

Table 1 and 2 have been removed. As suggested, the discussion on the earlier results has been reduced for easy readability; refer to pages 17-19.

---

## [Editor Report · Decision Letter 1]

27 Sep 2019

Nursing Students and Nurses’ Knowledge and Attitudes regarding Children’s Pain: A Comparative Cross-sectional study

PONE-D-19-18155R1

Dear Dr. Amponsah,

We are pleased to inform you that your manuscript has been judged scientifically suitable for publication and will be formally accepted for publication once it complies with all outstanding technical requirements.

With kind regards,

Sharon Mary Brownie

Academic Editor

PLOS ONE

Additional Editor Comments (optional):

Responses have effectively addressed recommendations from reviewers
---

## [Editor Report · Acceptance letter]

2 Oct 2019

PONE-D-19-18155R1 

Nursing Students and Nurses’ Knowledge and Attitudes regarding Children’s Pain: A Comparative Cross-sectional study 

Dear Dr. Kusi Amponsah:

I am pleased to inform you that your manuscript has been deemed suitable for publication in PLOS ONE. Congratulations! Your manuscript is now with our production department. 

With kind regards,

on behalf of

Professor Sharon Mary Brownie 

Academic Editor

PLOS ONE